# Novel Magnetic Field Modulation Concept Using Multiferroic Heterostructure for Magnetoresistive Sensors

**DOI:** 10.3390/s20051440

**Published:** 2020-03-06

**Authors:** Long Pan, Mengchun Pan, Jiafei Hu, Yueguo Hu, Yulu Che, Yang Yu, Nan Wang, Weicheng Qiu, Peisen Li, Junping Peng, Jianzhong Jiang

**Affiliations:** 1College of Intelligence Science and Technology, National University of Defense Technology, Changsha 410073, Hunan, China; plong_2017@163.com (L.P.); pmc_nudt@vip.163.com (M.P.); garfield_nudt@163.com (J.H.); huyueguo1991@163.com (Y.H.); cheyulubill@163.com (Y.C.); yuyang0316@hotmail.com (Y.Y.); 2008qiuweicheng@sina.com (W.Q.); lips13@163.com (P.L.); 2School of Materials Science and Engineering, Zhejiang University, Hangzhou 310027, Zhejiang, China; wangnan0917@zju.edu.cn (N.W.); jiangjz@zju.edu.cn (J.J.)

**Keywords:** MR magnetic sensors, suppress 1/f noise, ferroelectric/ferromagnetic multiferroic heterostructure, equivalent magnetic circuit model

## Abstract

The low frequency magnetic field detection ability of magnetoresistive (MR)sensor is seriously affected by 1/f noise. At present, the method to suppress the influence of low frequency noise is mainly to modulate the measured magnetic field by mechanical resonance. In this paper, a novel modulation concept employing a magnetoelectric coupling effect is proposed. A design method of modulation structure based on an equivalent magnetic circuit model (EMCM) and a single domain model of in-plane moment was established. An EMCM was established to examine the relationship between the permeability of flux modulation film (FMF) and modulation efficiency, which was further verified through a finite element simulation model (FESM). Then, the permeability modulated by the voltage of a ferroelectric/ferromagnetic (FE/FM) multiferroic heterostructure was theoretically studied. Combining these studies, the modulation structure and the material were further optimized, and a FeSiBPC/PMN-PT sample was prepared. Experimental results show that the actual magnetic susceptibility modulation ability of FeSiBPC/PMN-PT reached 150 times, and is in good agreement with the theoretical prediction. A theoretical modulation efficiency higher than 73% driven by a voltage of 10 V in FeSiBPC/PMN-PT can be obtained. These studies show a new concept for magnetoelectric coupling application, and establish a new method for magnetic field modulation with a multiferroic heterostructure.

## 1. Introduction

Owing to the attractive advantages of high sensitivity, small size, and low power dissipation, magnetoresistive (MR) sensors are widely used in magnetic field sensing, such as current sensing, force sensing, and ferromagnetic target detection [1,2,3]. In particular, the giant magnetoresistive (GMR) sensors and tunneling magnetoresistive (TMR) sensors show their detection ability on the order of pico-Tesla (10^−12^ T) at hundreds of kHz frequency [4]. Unfortunately, MR sensors suffer from the problem that the detection ability normally drops to the level of nano-Tesla (10^−9^ T) at low frequency due to the 1/f noise. Therefore, 1/f noise reduction has become an important challenge for MR sensors with pico-Tesla resolution.

Up to now, several schemes have been proposed to reduce the MR 1/f noise via modulating the detected low frequency magnetic field to a high frequency output signal, including magnetic flux modulation, chopping technology, etc [5,6]. Among them, magnetic flux modulation has attracted the most interest and was successively put forward by many groups [6,7,8] for its advantage of effective suppression for both electric 1/f noise and magnetic 1/f noise. For example, Edelstein et al. proposed a magnetic flux modulation adopting the mechanical resonance with MEMS technology [9], in which a pair of flux concentrators integrated into MEMS driven by an electrostatic comb modulated the detected magnetic field to a higher frequency (tens of kHz). Thus, the 1/f noise of a spin-valve GMR element was reduced by hundreds or even thousands of times. However, this solution still has the disadvantage of poor modulation efficiency. For improving the modulation efficiency, vertical motion flux modulation (VMFM) has been recommended by Hu et al [10]. In their scheme, the magnetic flux modulation was realized by vibrating the flux modulation film up and down. Although the modulation efficiency reached 40%, the modulation frequency (about 7 kHz) was still lower than the 1/f noise’s knee frequency of TMR sensors, which limits further improvement in the magnetic field detection ability. 

Recently, many studies have shown that artificial ferroelectric/ferromagnetic (FE/FM) multiferroic materials can effectively regulate the permeability of FM by applying an electric field [11,12,13,14,15,16]. Lou et al. took advantage of the adjustable permeability of multiferroic materials, and realized electrostatically tunable inductors with multiferroic composite cores consisting of Metglas/lead zirconate titanate/Metglas [17]. Tkach et al. prepared Ni films on single crystalline Pb(Mg_1/3_Nb_2/3_)_0.7_Ti_0.3_O_3_ (PMN-PT) with (011) orientation, which achieved about a 10-fold reduction of permeability after loading a −4 kV/cm electric field [18]. Phuoc et al. further demonstrated that the FeCo (100 nm)/NiFe (5 nm) magnetic film prepared on PMN-PT with (011) orientation has stronger magnetic permeability regulation ability compared with that on PMN-PT with (001) and (111) orientation [19]. Their results showed that the permeability of the FM layer prepared on (011) substrate can be reduced from about 270 to 70 under an electric field of 6.4 kV/cm, which verifies the regulation ability of FE/FM multiferroic materials on the magnetic permeability. They also provided the good idea of modulating the measured magnetic field by applying an alternating-curent (AC) electric field on the FE substrate to periodically adjust the permeability of the FM modulation film.

In this paper, a novel modulation concept called magnetic flux electric modulation (MFEM) is proposed. In this new scheme, the measured magnetic field is modulated by altering magnetic susceptibility of the flux modulation film (FMF), which is realized by the magnetoelectric coupling effect of a FE/FM multiferroic heterostructure. In order to study the effect of magnetic susceptibility of the FMF on modulation efficiency, a modified equivalent magnetic circuit model was established, which showed that the modulation efficiency is mainly related to the magnetic susceptibility modulation ability of the FMFurthermore, the theoretical calculation formula of magnetic susceptibility modulation ability was derived by a single domain model of in-plane moment. According to the above theoretical analysis, FeSiBPC/PMN-PT was selected as the FE/FM material and the magnetic susceptibility modulation ability was measured after preparing the samples. The results show that the actual magnetic susceptibility modulation ability of FeSiBPC/PMN-PT is in good agreement with the theoretical prediction. This shows that the theory of magnetic susceptibility modulation ability is reliable, and it is expected that MFEM will improve the performance of the sensor.

## 2. Principle

The sketch map of a prototype magnetic sensor based on MFEM is shown in Figure 1a. The prototype magnetic sensor includes MR sensors, flux concentrators (FC), and the modulation structure which consists of a FE substrate and a flux modulation film (FMF). The MR sensor is used to measure magnetic fields, and would suffer from the problems of low sensitivity and 1/f noise if it worked alone. The flux concentrators (FC), consisting of two soft magnetic films on both sides of the MR sensor, are used to enhance the sensitivity of the MR magnetic sensor. The FMF is located above the air gap of the FC. Different to the VMFM structure prepared in MEMS actuators [7,10,20], the FMF in this new scheme is prepared in a FE substrate, which constructs an artificial FE/FM multiferroic heterostructure. Due to the magnetoelectric coupling, the permeability of the FMF can be regulated when the electric field is applied to the FE substrate [19,21,22]. A more detailed process of permeability regulation is described as follows: A driving voltage *V_cc_* is loaded on the FE/FM multiferroic heterostructure to produce a strong anisotropic strain in the FE layer, F.; then, this anisotropic strain is transferred to the FMF, which affects the permeability of the FM layer through a magnetostriction effect. In this case, the magnetic flux in the air gap of the MFEM scheme would change accordingly. For example, when *V_cc_* is equal to zero, due to high permeability of the FMF, the magnetic flux around the air gap mainly passes through the FMF and the magnetic flux via the air gap is rare, as schematically shown with a thin arrow in Figure 1b. On the other hand, when *V_cc_* increases to *V_i_* (a voltage value closed to the *V*_max_), the permeability of the FMF will decrease due to the larger magnetic anisotropy induced by the strain coupling, so the magnetic flux through the air gap is heightened, as schematically shown with a thick arrow in Figure 1c. The permeability of the FMF and magnetic flux through the air gap easily alter periodically at the same frequency as that of the driving voltage *V_cc_*, as illustrated in Figure 1d. Afterward, the MR element in the air gap can detect an alternating magnetic field signal. It is worth noting that the frequency of the alternating magnetic field signal in the air gap is identical with that of the driving voltage *V_cc_*. Two advantages for this situation are: 1) it is easy to accurately measure the magnetic field signal by phase lock technique, and 2) the magnetic field signal is modulated to a high frequency, which makes it more convenient and effective to restrain the 1/f noise merely by adding a high-pass filter. As a result, the low frequency detection ability of the MR sensors will be significantly improved.

## 3. Theoretical Model

### 3.1. Establishment of the Equivalent Magnetic Circuit Model (EMCM) 

For giving a description of the relationship between the FMF’s permeability modulated by the electric field and the modulation efficiency, an equivalent magnetic circuit model (EMCM) was established as shown in Figure 1e. In this model, the magnetic flux source *Φ* stems from the magnetic flux aggregation of the FC, and its value is determined by the size and permeability of the flux concentrators. Due to the existence of the air gap and FMF, the magnetic flux *Φ* passes through three paths in the vicinity of air gap: part 1 passes through the FMF, and the corresponding magnetic reluctance *R*_p1_ can be changed with the permeability of FMF; parts 2 and 3 pass through the air gap and the ambient space area of air gap, respectively, and the relevant magnetic reluctance is defined as *R*_p2_ and *R*_p3_. These three magnetic reluctances are connected in parallel with each other.

Similar to the electrical resistance calculation formula, the magnetic reluctance of a magnetic circuit can be expressed as follows [23]:(1)$$R=\frac{l}{\mu_{r}\mu_{0}A}$$
where *l* is the length of the magnetic circuit, *μ*_r_ is the relative permeability of the magnetic circuit, *μ_0_* is the permeability of vacuum, and *A* is the cross-sectional area of the magnetic circuit. Based on this model, the magnetic reluctance of two circuits (*p1* and *p2* as shown in Figure 1b) can be expressed as follows:(2)$$R_{p1}=\frac{l_{g}}{\mu_{F}\mu_{0}w_{fc}t_{m}}$$
(3)$$R_{p2}=\frac{l_{g}}{\mu_{0}w_{fc}t_{fc}}$$

As shown in Figure 1a, *w*_fc_*,* and *t*_fc_ represent width and thickness of FC, respectively, *l*_g_ represents length of gap, and *t*_m_ represents thickness of FMF. However, as *p3* of the magnetic flux passes through an open space, it is hard to obtain *R_p3_* by the Equation (1). Luckily, the relationship between magnetic reluctance *R*_p2_ and *R*_p3_ is fixed once the size and permeability of flux concentrators are confirmed. *R_p2_* and *R_p3_* have a simple proportional relationship as follows:(4)$$R_{p3}=kR_{p2}$$

When the size of FC is determined, the values of $R_{p2}$ and $R_{p3}$ are fixed. Therefore, *k* is a constant parameter decided by the size of the FC. It should be pointed out that the MR elements are located in the air gap in our system, the magnetic flux, *Φ_p2_*_,_ through the air gap, according to equivalent magnetic circuit model, can be expressed as:(5)$$\Phi_{p2}=\frac{t_{fc}}{\left(\left(k+1\right)/k\right)t_{fc}+\mu_{F}t_{m}}\Phi$$

The magnetic flux in the air gap Φ*_p2_* is controlled by the permeability of FMF $\mu_{F}$. When $\mu_{F}$ is modulated from 1 to $\mu_{F}^{b}$, the modulation efficiency can be evaluated using the following equation:(6)$$\eta_{b}=\frac{\Phi_{0}-\Phi_{p2}^{\mu_{F}^{b}}}{\Phi_{0}}=1-\frac{\left(\left(k+1\right)/k\right)t_{fc}+t_{m}}{\left(\left(k+1\right)/k\right)t_{fc}+\mu_{F}^{b}t_{m}}$$
where Φ_0_ is the magnetic flux in the air gap when the relative permeability of FMF $\mu_{F}$ is equal to 1. As discussed above, the driving alternating voltage modulates the $\mu_{F}$ between $\mu_{F}^{a}$ and $\mu_{F}^{b}$ (assuming that $\mu_{F}^{a}$<$\mu_{F}^{b}$) due to the influence of magnetoelectric coupling effect, thus Φ*_2_* correspondingly changes between $\Phi_{p2}^{\mu_{F}^{a}}$ and $\Phi_{p2}^{\mu_{F}^{b}}$ ($\Phi_{p2}^{\mu_{F}^{a}}$>$\Phi_{p2}^{\mu_{F}^{b}}$), resulting in the MR elements obtaining a correspondingly changed magnetic field.

In case the Equation (6) is validated, the arbitrary modulation efficiency $\eta_{ab}$ can be acquired by:(7)$$\eta_{ab}=\eta_{b}-\eta_{a}$$

The modulation efficiency of this system can be expressed by:(8)$$\eta_{ab}=\frac{\Phi_{p2}^{\mu_{F}^{a}}-\Phi_{p2}^{\mu_{F}^{b}}}{\Phi_{0}}$$

### 3.2. Verification of the EMCM

We further established a finite element simulation model (FESM) by adopting the AC/DC Module of COMSOL Multiphysics which solves the magnetic field problems by calculating Maxwell equations. Like the magnetic structure showed in Figure 1a, the FESM includes the flux concentrators, flux modulation film and the space which they are in. By computing the FESM, the relationship between modulation efficiency $\eta_{b}$ and parameters ($\mu_{F}^{b}$, *t_fc_* and *t_m_*) is established. Furthermore, the FESM also provides a method to obtain the unknown parameter *k* in the formula. Using the result acquired by computing the FESM and Equation (6), the parameter *k* can be calculated as shown in Equation (9). Theoretically, parameter *k* is affected by the size of the FC rather than the FMF’s thickness *t_m_* and the FMF’s permeability $\mu_{F}^{b}$. However, there is an error in the calculated value of parameter *k* via FESM. In order to improve the precision of *k* value, the average value of *k* is obtained under the conditions of different parameter values of the FMF’s thickness *t_m_* and the FMF’s permeability $\mu_{F}^{b}$ (we assumed that *t_m_* has *m* numbers and $\mu_{F}^{b}$ has *n* numbers). The calculation formula is:(9)$$k=\frac{1}{mn}\sum_{i}^{m}\sum_{j}^{n}\frac{-\eta_{b}t_{fc}}{t_{mi}-\mu_{Fj}t_{mi}+t_{fc}\eta_{0}+\mu_{Fj}t_{mi}\eta_{0}}$$

Meanwhile, to verify the correctness of the above model, the finite element simulation model in different situations was calculated. The related parameters and calculated *k* value according to the simulation data of FESM and Equation (9) are shown in Table 1.

The modulation efficiency of the EMCM was calculated by putting the parameter k into Equation (6), and the result compared with simulation data of the FESM as shown in Figure 2. Figure 2a–d represents the results of situations a, b, c, and d respectively, and the point diagrams represent the results of the FESM while the line diagrams represent the results of the EMCIt was found that FESM and EMCM have a great coincidence, which effectively verified the correctness of the equivalent magnetic circuit model and the reliability of the method for calculating the value of parameter *k*. On the other hand, in different situations a, b, c, or d, parameter k is different, which also shows that parameter *k* is related to the size of the FC.

All of the above discussion shows that the finite element simulation model is a significant tool to prove the correctness of the EMCM and calculate the value of parameter *k*. The EMCM model can describe our system and study the modulation efficiency changed by the permeability of the FMF. More importantly, this also provides a way to optimize the sensor structure and materials as follows.

## 4. The Method to Improve Modulation Efficiency

### 4.1. Size Optimization of Modulation Structure

According to the relationship between modulation efficiency and related parameters (*t_fc_*, *t_m,_* and $\mu_{F}$), the value of the modulation efficiency can be further improved by optimizing the parameter *t_m_*. By computing the derivative of the modulation efficiency with respect to *t_m_*, the following relationships can be gotten:(10)$$\begin{array}{l} \frac{\partial \eta_{ab}}{\partial t_{m}}>0,\;(0<t_{m}<\frac{\left(\left(k+1\right)/k\right)t_{fc}}{\sqrt{\left(\mu_{F}^{a}-1\right)\left(\mu_{F}^{b}-1\right)}-1})\\ \frac{\partial \eta_{ab}}{\partial t_{m}}=0,\;(t_{m}=\frac{\left(\left(k+1\right)/k\right)t_{fc}}{\sqrt{\left(\mu_{F}^{a}-1\right)\left(\mu_{F}^{b}-1\right)}-1})\\ \frac{\partial \eta_{ab}}{\partial t_{m}}<0,\;(t_{m}>\frac{\left(\left(k+1\right)/k\right)t_{fc}}{\sqrt{\left(\mu_{F}^{a}-1\right)\left(\mu_{F}^{b}-1\right)}-1}) \end{array}\}$$

It is easy to conclude that the modulation efficiency increases firstly and decreases later by increasing the thickness of FC (*t_m_*). This means that for $t_{m}=\left(\left(k+1\right)/k\right)t_{fc}/\left(\sqrt{\left(\mu_{F}^{a}-1\right)\left(\mu_{F}^{b}-1\right)}-1\right)$, the optimal value of the modulation efficiency can be obtained and expressed by: (11)$$\begin{array}{c} \eta_{ab\_op}=\frac{\sqrt{\left(\mu_{F}^{b}-1\right)/\left(\mu_{F}^{a}-1\right)}-1}{\sqrt{\left(\mu_{F}^{b}-1\right)/\left(\mu_{F}^{a}-1\right)}+1}\\ =\frac{\sqrt{\chi_{F}^{b}/\chi_{F}^{a}}-1}{\sqrt{\chi_{F}^{b}/\chi_{F}^{a}}+1} \end{array}$$

Equation (11) shows that the optimal modulation efficiency of MFEM is only affected by the FMF’s magnetic susceptibility modulation ability $\chi_{F}^{b}/\chi_{F}^{a}$. Figure 3a shows the relationship between modulation efficiency and the FMF’s magnetic susceptibility modulation ability. It is found that the modulation efficiency can be enhanced by enlarging the FMF’s magnetic susceptibility modulation ability $\chi_{F}^{b}/\chi_{F}^{a}$. On the other hand, as shown in Figure 3b, the growing rate of modulation efficiency drops as the $\chi_{F}^{b}/\chi_{F}^{a}$ increases. Hence, the overlarge design value $\chi_{F}^{b}/\chi_{F}^{a}$ is not a good choice considering that it is difficult to achieve and will cause greater resource consumption. When the value of $\chi_{F}^{b}/\chi_{F}^{a}$ is bigger than 9, the corresponding modulation efficiency is bigger than 50%, which is larger than previously reported values with VMFM [24,25,26].

### 4.2. Materials Optimization FE Layer and FMF Layer 

Based on the above analysis, the key to achieving greater modulation efficiency is to realize greater magnetic susceptibility modulation ability of the FMIn our modulation schematic, the control of the FMF’s magnetic susceptibility is realized via the magnetoelectric coupling effect of artificial FE/FM multiferroic heterostructure. Although Teach et al. reduced the relative magnetic permeability of the FM layer by about 10 times when the electric field increased from 0 to −4 kV/cm through Ni/PMN-PT (011) multiferroic heterostructures [18], how the susceptibility was tuned by the electric field is still not clear. Hence, a theoretical study of the electrical regulation of the susceptibility was put forward, for gaining further insight into the modulation mechanism of the FMF’s magnetic susceptibility and thereby guiding its design [27,28].

Because the FMF layer is a magnetic film, the direction of magnetic moment is mainly in the plane. In order to simplify the analysis, a single domain model of in-plane moment is established to analyze the regulation ability of the magnetic susceptibility, as shown in Figure 4a–b. The free energy of the FMF structure includes not only the anisotropy energy $F_{K}$, demagnetizing field energy $F_{d}$, and Zeeman energy $F_{zeeman}$, but also the stress energy $F_{stress}$ due to the magnetoelectric coupling. Particularly, because the magnetic moment is in the xy-plane, and anisotropic components perpendicular to the xy-plane do not affect the direction of the magnetic moment, the anisotropy energy $F_{K}$ only has to take into account the energy generated by the anisotropy $K_{u1}$ in the xy-plane. For enlarging stress energy $F_{stress}$, the stress loaded in the magnetic material should be anisotropic. Then the free energy can be calculated by:(12)$$F_{all}^{stress}=-K_{u1}\cos^{2}\alpha -\frac{3}{2}\lambda_{s}\left(\sigma_{yy}-\sigma_{xx}\right)\cos^{2}\gamma -\mu_{0}HM_{s}\sin\gamma +\frac{1}{2}\mu_{0}M_{s}^{2}\left(N_{1}\cos^{2}\gamma +N_{2}\sin^{2}\gamma \right)$$
where $\sigma_{xx}$ and $\sigma_{yy}$ are the stress along the direction of *x*-axis and *y*-axis, respectively, F.; $\lambda_{s}$ represents the magnetostriction coefficient of FMF; *M_s_* represents the saturation magnetization; α is the angle between magnetization and $K_{u1}$; and γ is the angle between stress $\sigma_{yy}$ and magnetization which changes along the direction of magnetization *M_s_*. $N_{1}$ and $N_{2}$ are demagnetization factors in the direction x axis and y axis, respectively. In order to avoid the influence of the demagnetization field on the modulation, we can design the FMF into a circular film, $N_{1}$ equals $N_{2}$. In this case, Equation (12) can be simplified to:(13)$$F_{all}^{stress}=-K_{u1}\cos^{2}\alpha -\frac{3}{2}\lambda_{s}\left(\sigma_{yy}-\sigma_{xx}\right)\cos^{2}\gamma -\mu_{0}HM_{s}\sin\gamma +\frac{1}{2}\mu_{0}N_{1}M_{s}^{2}$$

According to the S-W theory, the magnetization stabilizes at the state of minimal free energy. Next, two situations for *V*_cc_ = 0 and *V*_cc_ = *V*_i_ were considered.

When *V*_cc_ is equal to zero, the magnetoelastic anisotropy could be ignored. In this case, the direction of magnetization is between the anisotropy $K_{u1}$ and magnetic field *H*, as shown Figure 4b. The stable direction is decided by:(14)$$K_{u1}\sin\left(2\alpha \right)-\mu_{0}HM_{s}\cos\gamma =0$$
where the γ is the angle between anisotropy and y-axis, and it is a constant value when the material of FMF is confirmed. α is the angle between $K_{u1}$ and $M_{s}$. Because the application occasion of magnetic sensors is mainly in the geomagnetic environment, the anisotropic energy is much larger than the Zeeman energy, which leads to a tiny angle of α. At this situation, it is feasible that sinα≈α and cosα≈1. Then, the angle α can be expressed by:(15)$$\alpha \approx \frac{s}{2}\cos\gamma$$

The parameter *s* is a tiny value for the geomagnetic application environment of magnetic sensors and can be expressed as $s=\mu_{0}HM_{s}/K_{u1}$. Because the angle β is also a tiny value, the magnetic susceptibility calculation formula at the direction of magnetic field *H* is:(16)$$\chi_{F}^{b}=\left(\frac{\partial M_{s}\sin\left(\gamma +\alpha \right)}{\partial H}\right)$$

Then, according to Equation (15) and (16), it can be gotten:(17)$$\chi_{F}^{b}=\frac{\mu_{0}M_{s}^{2}}{2K_{u1}}\cos^{2}\gamma$$

When the magnetic material is in the polycrystalline state, the angle γ is random and thereby the average value of $\cos^{2}\gamma$ is equal to $\frac{1}{2}$. Then, the magnetic susceptibility of polycrystalline magnetic material without stress can be expressed by:(18)$$\chi_{F}^{b}=\frac{\mu_{0}M_{s}^{2}}{4K_{u1}}$$

When *V_cc_* is equal to the drive voltage *V_i_*, the direction of magnetization is between the anisotropy $K_{u1}$ and magnetic field *H*, as shown in Figure 4a. γ is the angle between anisotropy and y-axis, and it is a constant value when the material of FMF is confirmed. β is the angle between y-axis and $M_{s}$. $F_{stress}$ is usually much larger than $F_{K}$ and $F_{zeeman}$. Then, it can be considered that angle β is a tiny value, and can be expressed by:(19)$$\begin{array}{ll} \beta & =0.5\frac{m\sin2(\frac{\pi }{2}-\gamma )}{2-m\cos2(\frac{\pi }{2}-\gamma )}\\ & +\frac{1}{2}h+\frac{1}{4}mh\cos2(\frac{\pi }{2}-\gamma ) \end{array}$$
where h is equal to $\mu_{0}HM_{s}/\frac{3}{2}\lambda_{s}(\sigma_{yy}-\sigma_{xx})$, m is equal to $K_{u1}/\frac{3}{2}\lambda_{s}(\sigma_{yy}-\sigma_{xx})$. Similarly, the $\left(\frac{\pi }{2}-\gamma \right)$ is the angle between anisotropy and x-axis, and is a constant value when the material of FMF is confirmed. Because $F_{stress}$ is much larger than $F_{K}$ and the parameters h and m are tiny values, the magnetic susceptibility calculation formula at the direction of magnetic field *H* can be finally written as:(20)$$\begin{array}{ll} \chi_{F}^{a} & ={\left(\frac{\partial M_{s}\sin\beta }{H}\right)}_{H\to 0}\\ & =\frac{\mu_{0}M_{s}^{2}}{\frac{3}{2}\lambda_{s}(\sigma_{yy}-\sigma_{xx})}(\frac{1}{2}+\\ & \frac{1}{4}\frac{K_{u1}}{\frac{3}{2}\lambda_{s}(\sigma_{yy}-\sigma_{xx})}\cos2(\frac{\pi }{2}-\gamma )) \end{array}$$

When the magnetic material is polycrystalline, as the angle $\left(\frac{\pi }{2}-\gamma \right)$ is random, the average value of $\left(\frac{\pi }{2}-\gamma \right)$ is equal to 0. The magnetic susceptibility of polycrystalline magnetic material in the situation of loading stress can be expressed by:(21)$$\chi_{F}^{a}=\frac{\mu_{0}M_{s}^{2}}{3\lambda_{s}(\sigma_{yy}-\sigma_{xx})}$$

In our modulation model, the anisotropic stress $\left(\sigma_{yy}-\sigma_{xx}\right)$ stems from the inverse piezoelectric effect of the FE layer. The anisotropic stress can be calculated by:(22)$$\sigma_{yy}-\sigma_{xx}=\frac{Y_{FMF}\left(d_{32}-d_{31}\right)E}{\left(1-\nu \right)}$$

Where $Y_{FMF}$ is the Young modulus of FMF layer; ν is the Poisson’s ratio of FMF layer; $d_{32}$ and $d_{31}$ are piezoelectric coefficients of the FE layer. E represents the amplitude of the electric field applied at the FE layer. According to Equation (11), the modulation efficiency of MFEM is only affected by FMF’s magnetic susceptibility modulation ability $\chi_{F}^{b}/\chi_{F}^{a}$. Combining Equations (18), (21), and (22), the following expression can be deduced:(23)$$\frac{\chi_{F}^{b}}{\chi_{F}^{a}}=\frac{3}{4}\frac{\lambda_{s}Y_{FMF}}{K_{u1}\left(1-\nu \right)}(d_{32}-d_{31})E$$

The electric field *E* depends on the thickness of FE layer and voltage, then the magnetic susceptibility modulation ability can be expressed by:(24)$$\frac{\chi_{F}^{b}}{\chi_{F}^{a}}=\frac{3}{4}\frac{\lambda_{s}Y_{FMF}V_{i}}{K_{u1}d\left(1-\nu \right)}(d_{32}-d_{31})$$

Where $V_{i}$ represents the voltage loaded in the FE layer, and *d* represents the thickness of FE layer. From Equation (24), the factors affecting the magnetic susceptibility modulation ability include two parts: parameters of the FM layer which include $\lambda_{s}$, $Y_{FMF}$, and ν, as well as $K_{u1}$, and parameters of the FE layer which consist of $\left(d_{32}-d_{31}\right)$ and E. For the purpose of enhancing the magnetic susceptibility modulation ability of the FMF, on one hand, for the FM layer, those are all great choices to enlarge the magnetostriction coefficient $\lambda_{s}$, Young’s modulus *Y*, and diminish the anisotropy *K_u1_* and Poisson’s ratio ν. It is worth pointing out that the theory is still valid for amorphous materials because the direction of anisotropy *K_u1_* generated by internal stress is still random. In this case, the amorphous magnetic material usual with low anisotropy *K*_u1_ is more suitable. On the other hand, for the FE layer, it is also effective to boost the FMF’s magnetic susceptibility modulation ability by enlarging $\left(d_{32}-d_{31}\right)$ and E.

Combining the above results, iron-based amorphous magnetic material is a great choice as the FMF, for its tiny magnetic anisotropy ($K_{u1}=38$ J/m^3^ [28]) and large magnetostriction coefficient (about 27 ppm [29]). On the other hand, single crystalline piezoelectric substrate Pb(Mg_1/3_Nb_2/3_)_0.7_Ti_0.3_O_3_ (PMN-PT) with (011) orientation can be selected as the FE layer, because its anisotropic piezoelectric coefficient $\left(d_{32}-d_{31}\right)$ reaches about 2500 pC/N [30]. Therefore, in order to verify the correctness of the theory and the feasibility of magnetic susceptibility modulation, 100 nm thick iron-based amorphous magnetic material FeSiBPC was prepared on PMN-PT by magnetron sputtering. The M–H curves of the FeSiBPC under different electric fields are shown in Figure 4c. When no electric field is loaded in PMN-PT. According to the data in the figure, the actual magnetic susceptibility modulation ability reached 150 times and the saturation magnetization *M_s_* was 1.1 × 10^6^ A/m. According to the M–H curve at E = 0 kV/cm, the *H_k_* of the FeSiBPC is about 2 Oe. So, the anisotropy of FeSiBPC *K_u1_* = 1/2**M_s_***H_k_* = 110 J/m^3^ [31]. When electric field E is equal to 5 kV/cm, the *H_k_* increases to 200 Oe, then we can get $H_{K}^{strain}=198$ Oe. So, the magnetostriction coefficient $\lambda_{s}=\left(H_{K}^{strain}M_{s}\right)/\left(3Y\left(\varepsilon_{xx}-\varepsilon_{yy}\right)\right)\approx 53$ ppm [31] (assume Young modulus $Y_{FMF}$ and Poisson’s ratio ν are 110 GPa and 0.3, respectively [28]). Anisotropy Ku1 and magnetostriction coefficient λs are basically consistent with that reported of iron-based amorphous magnetic material. Meanwhile, according to Equation (24), the theoretical magnetic susceptibility modulation ability reached 71 times. It can be concluded that Equation (23) can well analyze the magnetic susceptibility modulation ability of the FM/FE multiferroic heterostructure. According to Equations (11) and (24), the influence of driving voltage on the modulation efficiency is displayed in Figure 4d. The modulation efficiency increases with the increasing driving voltage *V*_i_ and dropping thickness of the FE layer *d*. In practical application, the driving voltage should not be too large. When the driving voltage is 10 V, modulation efficiency reaches 53% at the situation of *d* = 200 μm. Furthermore, when d decreases to 100 μm or 50 μm, the modulation efficiency can reach 64% or 73%, respectively.

Meanwhile, according to the 1/f noise characteristics of the magnetic sensor [6], when measuring magnetic field from low frequency (assumed to be 1 Hz) modulated to high frequency (*f_m_*), the 1/f noise of the magnetic sensor is reduced to:(25)$${\mathrm{noise}}_{1/f}^{f_{m}}=\frac{1}{\sqrt{f_{m}}}{\mathrm{noise}}_{1/f}^{1Hz}$$

Where, ${\mathrm{noise}}_{1/f}^{f_{m}}$ and ${\mathrm{noise}}_{1/f}^{1Hz}$ are the 1/f noise of the magnetic sensor working at the frequency of *f_m_* and 1 Hz, respectively. Therefore, when the low frequency measured magnetic field is adjusted to a magnetic field with a frequency of *f_m_*, the 1/f noise is reduced by $1/\sqrt{f_{m}}$ times. Due to the influence of modulation efficiency, the sensitivity of the magnetic sensor with modulation structure can be expressed by:(26)$$S_{m}=\eta S$$

Where *S_m_* represents the sensitivity of the magnetic sensor with modulation structure, η represents modulation efficiency, and *S* represents the sensitivity of the magnetic sensor without modulation structure. On the other hand, the magnetic field detection ability of the MR sensor can be represented by equivalent magnetic noise:(27)$$N_{mag}=\frac{{\mathrm{noise}}_{1/f}}{S}$$

So, the magnetic field detection ability of the MR sensor with modulation structure can be represented by equivalent magnetic noise:(28)$$\begin{array}{ll} N_{mag}^{f_{m}} & =\frac{{\mathrm{noise}}_{1/f}^{f_{m}}}{\eta S}\\ & =\frac{1}{\sqrt{f_{m}}\eta }\frac{{\mathrm{noise}}_{1/f}^{1Hz}}{S}\\ & =\frac{1}{\sqrt{f_{m}}\eta }N_{mag} \end{array}$$

Where $N_{mag}^{f_{m}}$ is the equivalent magnetic noise of MR sensor with modulation structure, *η* is modulation efficiency, and *N_mag_* is the equivalent magnetic noise of MR sensor without modulation structure. According to Equation (28), the modulation frequency *f_m_* and modulation efficiency *η* should be increased to improve the magnetic field detection ability of the magnetic sensor with modulation structure.

In terms of the working feasibility at high frequency of the magnetic modulation structure, up to now, a series of work has demonstrated that the magnetism modulation in FE/FM multiferroic heterostructures can work well under the AC electric field. For example, Kuntal Ray et al. theoretically proved that the magnetization switching time of the single-domain nanomagnet under the driving voltage applied on the magnetostrictive/piezoelectric multiferroic can be up to 10 ns [32], which implies that the modulation frequency of the novel modulation method can theoretically reach 100 MHz. Experimentally, Zhaoqiang Chu et al. proposed a low-power and high-sensitivity magnetic field sensor based on a FE/FM multiferroic heterostructure, and the sensor operates well at high frequency up to 40 kHz [33]. Tianxiang Nan et al. reported on nanomechanical magnetoelectric antennas with a FE/FM thin-film multiferroic heterostructure working at about 60 MHz. These researchers all suggest that the novel modulation scheme using a FE/FM multiferroic heterostructure could work well at a high modulation frequency. On the other hand, by designing the modulation structure as a resonance structure, the AC magnetoelectric coupling effect of the FE/FM multiferroic heterostructure can be further improved [34], which is conducive to further enhancing the modulation efficiency and reducing the work voltage. This is because the same working voltage can generate greater strain under the resonant state, thus enhancing the magnetic susceptibility modulation ability of the FMMeanwhile, 1/f knee frequency of MR sensors can usually reach above 500 kHz or even MHz [35]. Therefore, the magnetic flux electric modulation method can work at a frequency higher than the 1/f knee frequency of MR sensors. So, if the modulation frequency is 500 kHz and the modulation efficiency reaches 73%, the magnetic field detection ability can be improved by 516 times.

## 5. Conclusions

In conclusion, the magnetic flux electric modulation can efficiently modulate the measured magnetic field in the air gap by magnetoelectric coupling effect. A design method was established to improve modulation efficiency by optimizing the size of the FMF and designing the material used in the FE/FM multiferroic heterostructure. In the meantime, this novel modulation method employing an artificial FE/FM multiferroic heterostructure works under stable mechanical conditions, which avoids the restraint of modulation frequency in the mechanical resonance modulation. Moreover, an equivalent magnetic circuit model was established to study the relationship between permeability of the FMF and modulation efficiency, which was further verified through the finite element simulation model. Subsequently, a theoretical model was established to study the magnetic susceptibility regulation by the voltage of the FE/FM multiferroic heterostructure. Our studies present the way to optimize the modulation structure and material. To verify all theoretical results, a FeSiBPC/PMN-PT sample was prepared by magnetron sputtering, and realized a magnetic susceptibility modulation ability of 104 times. Meanwhile, magnetic susceptibility regulation theory also was proved by the results of the FeSiBPC/PMN-PT sample. Finally, a theoretical modulation efficiency by a voltage of 10 V was verified in the FeSiBPC/PMN-PT structure. When the thickness of the PMN-PT is reduced to 50 μm, modulation efficiency can reach 73%, which could be further improved by reducing the driving voltage and FE layer’s thickness.

## Figures and Tables

**Figure 1 sensors-20-01440-f001:**
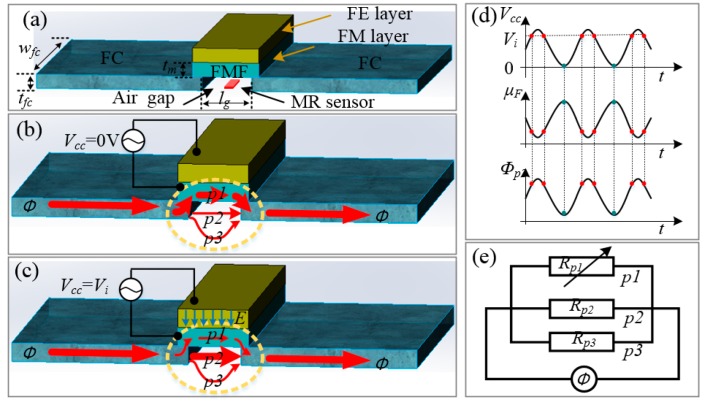
(**a**) Illustration of the prototype magnetic sensor: the schemata for magnetic flux through the three paths around the air gap for (**b**) *V_cc_* = 0 and (**c**) *V_cc_* = *V_i_* (a voltage value closed to the *V*_max_); (**d**) the schemata of the modulation principle: permeability of FMF *μ_F_* and magnetic flux through air gap *Φ_p2_* derived by an alternating-current (AC) voltage *V_cc_* loaded at FE layer; (**e**) equivalent magnetic circuit model. FE, ferroelectric; FM, ferromagnetic; FC, flux concentrator; FMF, flux modulation film; MR, magnetoresistive; *V_cc_*_,_ driving voltage; Rp, magnetic reluctance (parts 1, 2, and 3); *w*_fc_*,* and *t*_fc_ represent width and thickness of FC, respectively

**Figure 2 sensors-20-01440-f002:**
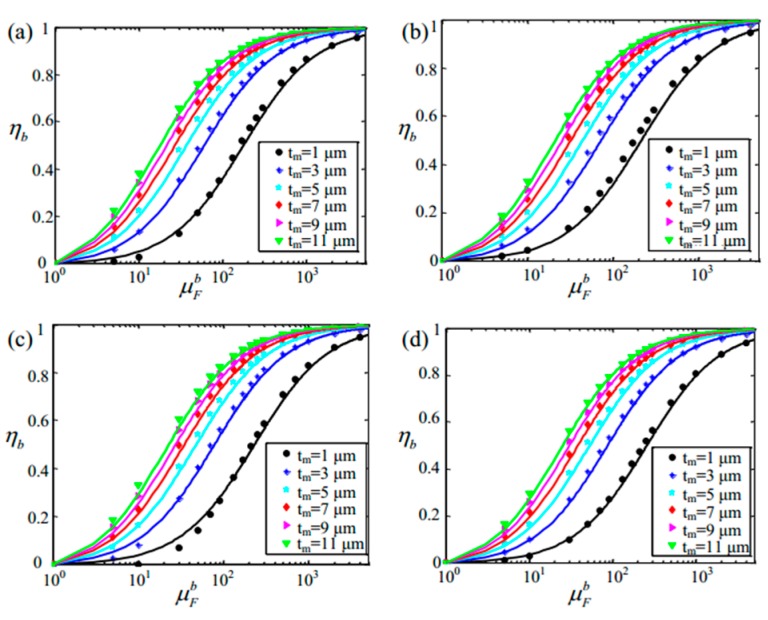
Modulation efficiency acquired by calculating the finite element simulation model (FESM) and the equivalent magnetic circuit model (EMCM) for situations (**a**–**d**), respectively. The points are the results of the FESM, and the line diagrams are the fitted results using the EMCM. $\eta_{b}$, modulation efficiency, F.; *t_m,_* FMF thickness.

**Figure 3 sensors-20-01440-f003:**
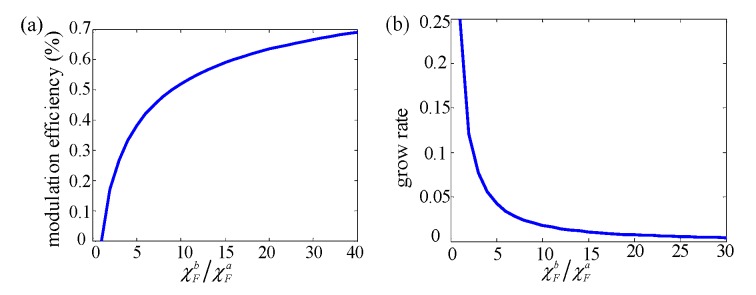
(**a**) The modulation efficiency as a function of the magnetic susceptibility modulation ability of FMF, M.; (**b**) the grow rate of the modulation efficiency as a function of the magnetic susceptibility modulation ability of FMF.

**Figure 4 sensors-20-01440-f004:**
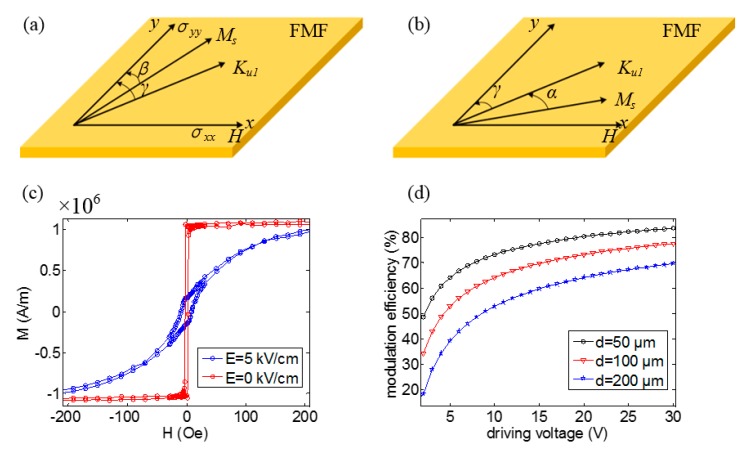
Illustration of magnetization direction under the impact of stress energy, magnetic crystal anisotropy energy, and Zeeman energy for (**a**) larger stress and (**b**) zero stress states, respectively.Thereinto, *M_s_* represents saturation magnetization. *K_u1_* represents anisotropy. *H* represents external magnetic field. *σ_xx_* and *σ_yy_* are the stress along the direction of *x*-axis and *y*-axis respectively. (**c**) the M–H curves of FeSiBPC/PMN-PT under different electric fields; (**d**) modulation efficiency as a function of driving voltage for FeSiBPC/PMN-PT under different FE thickness.

**Table 1 sensors-20-01440-t001:** Parameters in different situations.

Situations	*t_fc_* (μm)	*l_g_* (μm) (length of gap)	*w_fc_* (μm)	Permeability of FC	*k* (solved parameter)
a	5	30	300	500	2.92 × 10^−2^
b	10	30	300	500	4.94 × 10^−2^
c	5	40	300	500	2.21 × 10^−2^
d	10	40	300	500	4.07 × 10^−2^

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
