# Peer review of "Novel Magnetic Field Modulation Concept Using Multiferroic Heterostructure for Magnetoresistive Sensors"

_sensors, 2020, doi:10.3390/s20051440_

Round 1
Reviewer 1 Report
The authors present novel magnetic field modulation concept using multiferroic heterostructure for magnetoresistive (MR) magnetic sensors. It is known, that the low frequency magnetic field detection ability of MR magnetic sensors is seriously affected by the 1/f noise, therefore, the signal modulation possibilities with higher frequencies are intensively studied and reported in literature. The authors give summary of these works in the Introduction, explaining findings by other authors and existing problems. Therefore, the subject studied in this work is of great importance and the obtained results are interesting for the readers of MDPI Sensors journal.
However, in my opinion, the paper can not be published in the present form. I have the following comments:
1 presents ilustration of modulation principle and cases when Vcc=0 or Vi, however, it would be good also to present Vi in Fig.1 (d). As the arrow in the air gap is thick in this case, one can assume Vi close to Vmax..... The authors explain operation principle of the proposed Magnetic flux electric modulation (MFEM) in part 2. However, I didn‘t find any considerations about magnetic field values and ranges which could be measured by using this structure, dependence on film thickness and air gap dimensions. Also, as the sample of proposed structure was prepared, it would be good to present the possible magnetic field values, which could be measured by this method. The authors theoretically consider polycrystalline magnetic film, however, the prepared film is amorphous. Does it change the obtained properties? What would be difference if polycrystalline or epitaxial film would be used? The aim of this paper is to avoid disadvantages of 1/f noise at low frequency measurement range using higher frequency modulation. However, the discussion about possible modulation frequencies is only shortly given at the end of the paper citing other works. Also, it is not well explained why such material was used for testing modulation possibilities. Please explain more clearly the mentioned points. Also, the language of the paper has to be checked more carefully.
Author Response
----------------------------------------------------------------------
Report of the First Reviewer -- Sensors-722418
----------------------------------------------------------------------
The authors present novel magnetic field modulation concept using multiferroic heterostructure for magnetoresistive (MR) magnetic sensors. It is known, that the low frequency magnetic field detection ability of MR magnetic sensors is seriously affected by the 1/f noise, therefore, the signal modulation possibilities with higher frequencies are intensively studied and reported in literature. The authors give summary of these works in the Introduction, explaining findings by other authors and existing problems. Therefore, the subject studied in this work is of great importance and the obtained results are interesting for the readers of MDPI Sensors journal.
However, in my opinion, the paper cannot be published in the present form. I have the following comments:
Question 1: figure 1 presents illustration of modulation principle and cases when Vcc=0 or Vi, however, it would be good also to present Vi in Fig.1 (d). As the arrow in the air gap is thick in this case, one can assume Vi close to Vmax.....
Reply: Many thanks for reviewer’s great suggestion. We have already added Vi in Figure 1(d) of the manuscript.
Question 2: The authors explain operation principle of the proposed Magnetic flux electric modulation (MFEM) in part 2. However, I didn’t find any considerations about magnetic field values and ranges which could be measured by using this structure, dependence on film thickness and air gap dimensions. Also, as the sample of proposed structure was prepared, it would be good to present the possible magnetic field values, which could be measured by this method.
Reply: Since the magnetic field measurement can only be achieved by adding an MR sensor into the air gap, the measured magnetic field values and ranges are mainly determined by the MR sensor and the magnification of the flux concentrators. The description has been modified on line 88-92 and Figure 1(a) in the manuscript. However, in this work we focus on the concept of MFEM and its high modulation efficiency. In other words, the main purpose of this manuscript is to realize the magnetic field modulation concept by a novel method, so as to suppress the 1/f noise of MR sensors and improve the magnetic field detection ability. Therefore, we only prepared samples to prove the feasibility of realizing the high modulation efficiency with this novel magnetic field modulation method. As the reviewer points out, the magnetic field value and the range dependence on film thickness and air gap dimensions of the flux concentrators are not provided. This is because this manuscript mainly proposed the concept of modulation efficiency and focused modulation performance. According to equation (11) in the manuscript, the modulation efficiency has nothing to do with film thickness and air gap dimensions of the flux concentrators. Furthermore, there have been many studies on the relationship between the magnetic field measurement ability and sizes of the flux concentrators (References: IEEE Transactions on Magnetics, vol. 48, no. 4, April 2012; Sensors and Actuators A 97-98 (2002) 10-14; Sensors and Actuators A 142 (2008) 503-510.). So we do not have to study these in this manuscript. Even so, we have added the possible modulation efficiency for the structure and the influence on the sensitivity of the MR sensor in the revised version. Moreover, the MR sensor has also been added in the Figure 1(a) in the revised version for more detail.
Question 3: The authors theoretically consider polycrystalline magnetic film, however, the prepared film is amorphous. Does it change the obtained properties? What would be difference if polycrystalline or epitaxial film would be used?
Reply: We theoretically considered the more complex polycrystalline case (direction of anisotropy Ku1 is random), and according the theory, Ku1 needs to be reduced to increase the FMF’s magnetic susceptibility modulation ability. So, using the amorphous material, the magnetic crystal anisotropy can be further inhibited. This is significant for the modulation efficiency. On the other hand, it is inevitable for the amorphous material to have internal random stress, which can also lead to the randomness of effective anisotropy Ku1. As a result, all the discussions of our theory are still valid to amorphous materials. The explanation has been added at line 312-314 of the manuscript.
According to the theory and discussion, it is easy to conclude that changing the magnetic materials to polycrystalline or epitaxial film, the modulation efficiency would be reduced to some extent due to their bigger anisotropy Ku1.
Question 4: The aim of this paper is to avoid disadvantages of 1/f noise at low frequency measurement range using higher frequency modulation. However, the discussion about possible modulation frequencies is only shortly given at the end of the paper citing other works. Also, it is not well explained why such material was used for testing modulation possibilities. Please explain more clearly the mentioned points. Also, the language of the paper has to be checked more carefully.
Reply: Thanks for the reviewer’s valuable comments. To measure the modulation effect at a high frequency, a complete sensor should be made. We are preparing magnetic sensors but there are some problems in the preparation of the sensor. Considering the main purpose of this manuscript is to realize the magnetic field modulation concept by a novel method, we cited other researchers’ papers, in which the FE/FM multiferroic heterostructure can work at high frequencies up to MHz (up to the knee frequency of the MTJ sensors), to prove the feasibility of the modulation method. In the revised version, we provide more direct evidences that the FE/FM multiferroic heterostructure can work at a high frequency. Detailed instructions are added at line 365-381 of the manuscript.
For the material choice, we considered the following two aspects:
(a) PMN-PT was selected as the ferroelectric substrate. PMN-PT has a big piezoelectric coefficient, which is nearly an order of magnitude larger than that of the PZT. And its anisotropic piezoelectric coefficient (d31-d32) reaches to about 2500 pC/N.
(b) FeSiBPC was selected as the ferromagnetic material. According to equation (11) and equation (22) in the manuscript, decreasing anisotropy Ku1 and increasing magnetostriction coefficient λs can improve the modulation efficiency. The iron-based amorphous magnetic material is a good choice for FMF because it has tiny Ku1 and large λs (Ku1=38 J/m3 and λs=27 ppm (reference [28] and [29] in the manuscript)). So,we chose FeSiBPC as magnetic material of FMF. Through the estimation of the test data, we obtained that Ku1=110 J/m3 and λs=53 ppm of FeSiBPC, which meets our requirements. An analysis of the material properties of FeSiBPC is added in line 327-334 of the manuscript.
We have revised the language of the manuscript and marked the modified part in the revised version.

Reviewer 2 Report
The manuscript "Novel magnetic field modulation concept using multiferroic heterostructure for MR magnetic sensors" has been reviewed. This work proposes a high-efficiency magnetic flux modulation design, which serves for solving the low-frequency noise problem of magnetoresistance sensor. Comprehensive theoretical analyses including analytical models and finite element simulations are performed to assess the modulation efficiency of the design. Experiments further supports the practical viability of the design.
I think the quality of the paper meets the criteria of publication in Sensors for its novelty and importance. I have a few questions and suggestions as follows for the authors to further improve the paper.
1. More details need to be provided for the finite elment model. Which equations are actually solved? What software is used?
2. In Eq. 9, what is the reason for doing a summation? What are m and n?
3. In the stress modulation model, the author states that "the (gamma - alpha) is the angle between anisotropy and y-axis, and is a constant value when the material of FMF is confirmed". I suggest the authors introduce a notation for the quantity of (gamma - alpha) and then keeping either one of gamma or alpha without introducing the other, so it is clear that the formulations contain only one independent variable to be determined. Same can be applied to the case under voltage and stress, where gamma + alpha is the direction of anistropy axis.
4. In Fig. 4, since the direction of Ku is independent of H, it will help to draw Ku1 in Fig. 4(a) and that in 4(b) along the same direction, while drawing M_S along different directions.
5. In Eq. 18, K'_u1. Why the prime symbol?
6. A few typos I have caught:
Fig. 1(d), middle panel, the label "mu_MFM". Should be mu_F.
After Eqs. 2-3, "t_fc ... thickness of FMF" "t_m represents thickness of FC". Should be the other way around.
Section 4.2, "The M-H curves of FeSiBPC under different electric fields are shown in Figure 3(c)". Should be 4(c).
After Eq. 23, "noise^{f_m}_1/f is the 1/f noise of magnetic sensors when the working frequency is 1 Hz". Should be noise^{1Hz}_1/f.
Author Response
----------------------------------------------------------------------
Report of the Second Reviewer -- Sensors-722418
----------------------------------------------------------------------
The manuscript "Novel magnetic field modulation concept using multiferroic heterostructure for MR magnetic sensors" has been reviewed. This work proposes a high-efficiency magnetic flux modulation design, which serves for solving the low-frequency noise problem of magnetoresistance sensor. Comprehensive theoretical analyses including analytical models and finite element simulations are performed to assess the modulation efficiency of the design. Experiments further supports the practical viability of the design.
I think the quality of the paper meets the criteria of publication in Sensors for its novelty and importance. I have a few questions and suggestions as follows for the authors to further improve the paper.
Question 1: More details need to be provided for the finite element model. Which equations are actually solved? What software is used?
Reply: We used the AC/DC module of COMSOL Multiphysics software to establish the finite element simulation model and the equations calculated in this model are Maxwell’s equations. We have added the relative description in line 166-168 of the revised version. But the specific Maxwell’s equations were not presented because they are widely known.
Question 2: In Eq. 9, what is the reason for doing a summation? What are m and n?
Reply: In Eq. 9 in the manuscript, doing a summation is to calculate the average value of parameter k for improving its precision. A series of values of k were obtained using the finite element simulation under the conditions of different parameters of FMF’s thickness tm and FMF’s permeability . We used m, n to note the parameter series of tm and . Coefficient 1/nm was lost in Eq. 9 of the manuscript of previous version and has been added in the revised version. Line 175-178 in the revised version is modified.
Question 3: In the stress modulation model, the author states that "the (gamma - alpha) is the angle between anisotropy and y-axis, and is a constant value when the material of FMF is confirmed". I suggest the authors introduce a notation for the quantity of (gamma - alpha) and then keeping either one of gamma or alpha without introducing the other, so it is clear that the formulations contain only one independent variable to be determined. Same can be applied to the case under voltage and stress, where gamma + alpha is the direction of anisotropy axis.
In Fig. 4, since the direction of Ku is independent of H, it will help to draw Ku1 in Fig. 4(a) and that in 4(b) along the same direction, while drawing M_S along different directions.
Reply: Many thanks for reviewer’s valuable suggestion. Modifications have been made in these places of the revised version by changing the definition of the angle in Figure 4(a) and Figure 4(b) of the manuscript. In Figure 4(a) and Figure 4(b) of the manuscript, γ is defined as the angle between the anisotropy axis and y-axis, and it is a constant value when the material of FMF is confirmed. The β is defined as the angle between y-axis and Ms direction, while α is the angle between Ku1 and Ms. The relative changes of Figure 4(a) and Figure 4(b) in the manuscript have been redrawn according to the suggestions of reviewer. Line 264-265, 278-283 of the revision is also modified accordingly.
Question 4: In Eq. 18, K'_u1. Why the prime symbol?
Reply: should be Ku1. We have modified equation (18) in the manuscript by replacing with Ku1 in line 289 of the revised version.
Question 5: A few typos I have caught: Fig. 1(d), middle panel, the label "mu_MFM". Should be mu_F. After Eqs. 2-3, "t_fc ... thickness of FMF" "t_m represents thickness of FC". Should be the other way around. Section 4.2, "The M-H curves of FeSiBPC under different electric fields are shown in Figure 3(c)". Should be 4(c). After Eq. 23, "noise^{f_m}_1/f is the 1/f noise of magnetic sensors when the working frequency is 1 Hz". Should be noise^{1Hz}_1/f.
Reply: Thanks for your careful reading, and we have corrected the errors in these places. The corresponding modifies are in line 116, 141-142, 262, 346-347 of the revised version.

Reviewer 3 Report
The paper represents a really interesting concept. The magnetoelectric devices present a new era in field monitoring with much improved sensitivity. It is expected that flux concentration will improve the characteristics of the device.
The missing part of the paper is experiments. Authors need to add experiments on the dependence of sensitivity on frequency and results on the uncertainty of measurements.
If authors provide this information, I believe that the paper will gain high visibility.
Author Response
----------------------------------------------------------------------
Report of the third Reviewer -- Sensors-722418
----------------------------------------------------------------------
The paper represents a really interesting concept. The magnetoelectric devices present a new era in field monitoring with much improved sensitivity. It is expected that flux concentration will improve the characteristics of the device.
Question: The missing part of the paper is experiments. Authors need to add experiments on the dependence of sensitivity on frequency and results on the uncertainty of measurements.
If authors provide this information, I believe that the paper will gain high visibility.
Reply: Many thanks to the reviewer for the positive comments on our work. Exactly as the reviewer said, the frequency experimental result will further increase the visibility of our work. However, there are difficulties in the experiment of magnetism controlled by electric field with AC technology. For example, the typical electric field control of magnetism measured with SQUID only can obtain the DC magnetization response to the applied voltage, so it is hard to measure the permeability of FE/FM multiferroic heterostructure under a high-frequency voltage. The effective method of measuring the modulation effect at a high frequency is using a complete sensor as we proposed. We are preparing magnetic sensors to measure modulation performances at high frequency. However, there are some problems in the preparation technology of the sensor. Considering the main purpose of this manuscript is to realize the magnetic field modulation concept by a novel method, we cited other researchers’ papers, in which the FE/FM multiferroic heterostructure can work at high frequencies up to MHz (up to the knee frequency of the MTJ sensors), to prove the feasibility of the modulation method. The corresponding modifies in the revised version are in line 346-381 of the revised version. We believe the concept in the present manuscript is complete and it is expected to improve the characteristics of the device. Similarly, experiment results on the uncertainty of measurements also requires a complete sensor to be prepared. But, we theoretically analyze effects of the novel modulation method on noise and magnetic field detection ability of magnetic sensor.

Round 2
Reviewer 3 Report
The paper can be published in its present version